# PhysCtrl: Generative Physics for Controllable and Physics-Grounded Video Generation

**Chen Wang**[1][*], **Chuhao Chen**[1][*], **Yiming Huang**[1], **Zhiyang Dou**[2]
**Yuan Liu**[3], **Jiatao Gu**[1], **Lingjie Liu**[1]
[1]University of Pennsylvania, [2]MIT, [3]HKUST    [*] equal contribution
{chenw30,chuhaoc,ymhuang9,jgu32,lingjie.liu}@seas.upenn.edu
frankdou@mit.edu; yuanly@ust.hk
https://cwchenwang.github.io/physctrl

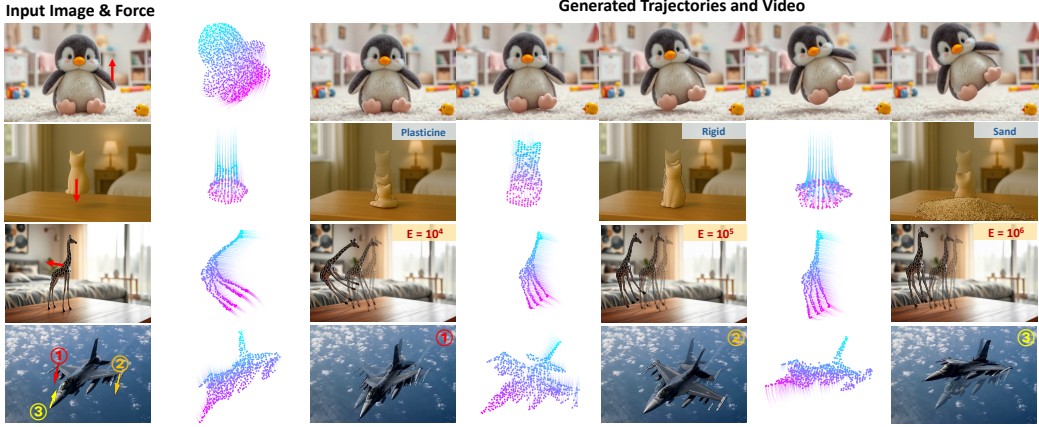

Figure 1: We propose PhysCtrl, a novel framework for physics-grounded image-to-video generation with physical material and force control. PhysCtrl supports generating physics-plausible motion trajectories across multiple materials as control signals (second row), and allows controls over physics parameters (e.g., **Young's Modulus** $E$ of elastic material (third row)) and **force** (last row). Note that in the bottom three rows, overlaid trajectories and frames use lighter hues for earlier time steps and darker hues for later ones.

## Abstract

Existing video generation models excel at producing photo-realistic videos from text or images, but often lack physical plausibility and 3D controllability. To overcome these limitations, we introduce PhysCtrl, a novel framework for physics-grounded image-to-video generation with physical parameters and force control. At its core is a generative physics network that learns the distribution of physical dynamics across four materials (elastic, sand, plasticine, and rigid) via a diffusion model conditioned on physics parameters and applied forces. We represent physical dynamics as 3D point trajectories and train on a large-scale synthetic dataset of 550K animations generated by physics simulators. We enhance the diffusion model with a novel spatiotemporal attention block that emulates particle interactions and incorporates physics-based constraints during training to enforce physical plausibility. Experiments show that PhysCtrl generates realistic, physics-grounded motion trajectories which, when used to drive image-to-video models, yield high-fidelity, controllable videos that outperform existing methods in both visual quality and physical plausibility.

39th Conference on Neural Information Processing Systems (NeurIPS 2025).

# 1 Introduction

Video generation has emerged as a transformative technology, powering applications in gaming [7, 80, 14], animation [10, 92, 26], autonomous driving [84, 86], digital avatars [30, 101], robotics [49]. Modern video generative models [62, 92, 94, 4] can produce photo-realistic videos from text or single images. However, they often lack physical plausibility, controllability over dynamic physical behaviors and high fidelity, because they are trained on massive 2D videos in a pure data-driven manner [2, 3].

To achieve physics-grounded video generation, incorporating inductive biases of physical dynamics is crucial. Driven by this, recent works have combined physics simulators [37, 1, 56] with neural representations (e.g., Gaussian splats) to simulate rigid or non-rigid dynamics and render them into videos [91, 38, 51, 8, 76] under scene-specific settings. While physics simulators based on Newtonian mechanics can model the dynamics of diverse real-world systems—including soft/rigid bodies, fluids, and gases [37, 59, 56], they suffer from high computational cost, sensitivity to hyperparameters (e.g., simulation substeps, grid size), numerical instabilities, and trade-offs between generality and accuracy. As a result, when directly using a physics simulator for video generation, people have to tune several hyperparameters and might need to switch simulators with regard to object material (*e.g*, MPM for elastic and rigid body simulators for rigid). It might also lack robustness and suffer from slow speed (especially for inverse problems).

To address these issues, we propose PhysCtrl, a framework for physics-grounded image-to-video generation with explicit control over physical parameters and external forces. A key component of our framework is a generative physics network, a diffusion-based model that learns the distribution of physical dynamics. It works on various material types, requires minimal user input and supports fast forward and backward. Conditioned on physical parameters and applied forces, it predicts physical dynamics that serve as control signals for pretrained video generative models [24]. In our design, we address two fundamental questions to achieve robust, efficient, and generalizable physics priors for controllable video generation:

*1. What is an appropriate representation of physical dynamics for providing control in video models?* We seek a representation that enables efficient control of video models while generalizing across a wide range of materials. Very recent work on controllable video generation [21, 24] has shown that video models can synthesize rich and coherent content from only sparse and explicit point controls. Meanwhile, point clouds offer greater flexibility and generalization for modeling different materials than other explicit representations, such as meshes or voxel grids, making them more suitable for learning-based generative physics networks. Considering these two aspects, we propose to represent physical dynamics as 3D point trajectories, enabling compact motion encoding and seamless integration with video generative models while supporting diverse material types.

*2. How to embed generative physics priors across various materials into a network?* High-quality and diverse data are essential for learning the distribution of physical dynamics (i.e., *generative physics*). We therefore collect a large-scale synthetic dataset of 550K object animations across four material types (elastic, sand, plasticine, and rigid), capturing complex, physics-grounded dynamics via physics simulators. Using this dataset, we design a diffusion model to generate physics-plausible 3D motion trajectories conditioned on physical conditions. Inspired by particle dynamics [37], where particles interact with neighbors to determine their next state, we introduce a novel spatiotemporal attention block in the diffusion model to emulate these interactions: it first aggregates spatial influences from neighboring points and then predicts each point's trajectory over time. Finally, to embed explicit physical knowledge directly into the network, we incorporate physics-based constraints during training, ensuring that the generated motions are physics-plausible.

We conduct comprehensive evaluations of our method, demonstrating our model can produce physics-plausible motion trajectories. We further show that the generated trajectories can be used as the input for a trajectory-conditioned video model for image-to-video generation, outperforming existing video generative models in both visual fidelity and physics plausibility. Our key contributions are:

- We introduce PhysCtrl, a novel and scalable framework that represents physics dynamics as 3D point trajectories over time, enabling physics-grounded image-to-video generation with explicit control over physical parameters and external forces.

- We develop a diffusion-based point trajectory generative model equipped with a spatiotemporal attention mechanism and physics-based constraints, efficiently learning generative physical dynamics across four material types.

- We collect a large-scale synthetic dataset of 550K object animations, spanning elastic, sand, plasticine, and rigid materials, using physics simulators. We will release this dataset to support future research in physical dynamics learning.

- We demonstrate the effectiveness of PhysCtrl in generating realistic, physics-grounded dynamics and achieve high-quality image-to-video generation results given user-specified physics parameters and external forces.

## 2  Related Work

**Neural Physical Dynamics**  Traditionally, physical dynamics are solved with numerical methods such as finite element method (FEM) [102], position-based dynamics (PBD) [60, 55], material point method (MPM) [37], smoothed-particle hydrodynamics (SPH) [17, 63, 43] and mass-spring systems [52]. Physical Informed Neural Networks (PINNs) [64] use neural networks to approximate the solution of partial differential equations and incorporate physics constraints in the loss functions. Combined with neural fields [58], PINNs achieve success in domains like fluids [13, 85] but are limited in per-scene optimization setting. Concurrent work, ElastoGen [19], replaces part of the physics simulation with neural networks for faster inference, but relies on a voxel representation, supports only elastic materials, and requires a full 3D model as input. Graph Neural Networks (GNNs) have emerged as an effective tool for modeling particle interactions with diverse material types [69, 93, 70, 95]. However, such approaches typically rely on next-step predictions for modeling dynamics, making them susceptible to drift and error accumulation over time. In contrast, our method represents objects as flexible point clouds and leverages a spatio-temporal trajectory diffusion model to robustly capture the dynamics of diverse materials in a unified framework.

**Controllable Video Generative Models**  Video generative models are trained on massive text-video paired datasets and achieve high-quality video generation [29, 4, 41, 11, 94]. Existing works have shown that additional control signals can be injected into pretrained models for controllable video generation, such as camera movement [25, 20], human pose [30], and point movement [21, 24, 5]. However, these models lack an understanding of physical laws and thus generate outputs that are often not physically plausible. Furthermore, they cannot support explicit physics control. Our work focuses on generating physics-grounded dynamics that can be used as a physics control signal for video models.

**Physics-Grounded Video Generation**  Existing methods leverage physics simulators to produce physics-grounded videos. One approach reconstructs neural representations from multi-view images, applies simulation on these representations, and then renders the results into video. For example, PhysGaussian [91], Spring-Gaus [99], and Vid2Sim [9] integrate MPM, spring–mass systems, and LBS-based simulation [59] into 3D Gaussians for simulation and rendering. VR-GS [38] applies physics-aware Gaussian Splatting in VR/MR for real-time, intuitive 3D interaction and physics-based editing. PhysDreamer [97] distills motions from video models to estimate physics parameters. These methods are scene-specific and require high-quality 3D reconstruction to achieve good results. Recently, researchers started to combine physics simulators with video generative models. PhysGen [51], PhysGen [8] and PhysMotion [76] generate videos of 2D rigid body dynamics or deformable dynamics. These methods rely on physics simulators to generate dynamics and coarse texture and only use video models for texture refinement. PhysAnimator [90] combines physical simulators and a sketch-guided video diffusion model for animations. Compared with methods that rely on physics simulators, our method embeds physics priors into a diffusion model, which avoids manual hyperparameter tuning and improves numerical stability for dynamics prediction. The predicted dynamics can be used as guidance for video generative models to synthesize physics-grounded and controllable videos. Concurrent works WonderPlay [48] and Force Prompting [22] also investigate using force as the condition signal for video generation.

**4D Dynamics**  Parametric models have been widely used to represent category-specific deformable shapes, such as SMPL and SMAL [54, 103] for human and animal bodies, FLAME [47] for faces, MANO [67] for hands. Recent advances in 4D dynamics have been exploring to capture object dynamics of arbitrary topologies [61, 57, 77, 45, 77, 12] with Neural-ODE and coordinate-MLPs.

With the success of diffusion models [28, 72, 73, 74] on high-quality generation on several modalities, including text [23], image [66, 68], audio [42, 44, 33], video [29, 27] and 3D [50, 71, 96, 53], researchers have started to learn the distribution of object dynamics with diffusion models [18, 6, 98, 88]. Motion2VecSets [6] introduced a 4D representation with latent vector sets, and trained a conditional diffusion model for dynamic reconstruction from sparse point cloud sequences. DNF [98] leverages a dictionary-based neural field to learn a compact motion space for unconditional 4D generation. However, these methods are only trained on datasets with a limited number of shapes that contain only human and animal motions, while our method focuses on learning physics-grounded dynamics, which contain a large variety of dynamic phenomena. We also use a more flexible point representation that is better suited for downstream tasks.

## 3 Preliminary

We generate ground-truth point trajectories for training our generative physics network (also referred to as "physics-grounded trajectory generative model") on data synthesized by physics simulators, including MPM and rigid body simulators. Here we review the basics of MPM, which form the basis for our physics-aware constraint in Section 4.

**Material Point Method** Material Point Method (MPM) [75, 65, 40, 37, 35, 31, 91] simulates the deformation of discrete material particles under the assumption of continuum mechanics, where the transformation of each particle from the material space to the world space is defined by a deformation mapping $\mathbf{x} = \phi(\mathbf{X}, t)$, and the associated deformation gradient $\mathbf{F} = \nabla_{\mathbf{X}}\phi(\mathbf{X}, t)$ measures the local deformation of the material such as rotation and stretch. The evolution of $\phi$ at time $t$ is governed by the conservation of mass and momentum, which can be formulated as

$$\rho\frac{D\mathbf{v}}{Dt} = \nabla \cdot \boldsymbol{\sigma} + \mathbf{f}_{ext} \qquad \frac{D\rho}{Dt} + \rho\nabla \cdot \mathbf{v} = 0 \tag{1}$$

where $\rho$, $\mathbf{v}$ and $\mathbf{f}_{ext}$ denote the density, the velocity field and the per-unit volume external force respectively. The Cauchy stress $\boldsymbol{\sigma} = \frac{1}{\det(\mathbf{F})}\frac{\partial\Psi}{\partial\mathbf{F}}(\mathbf{F})\mathbf{F}^{\top}$ and the energy density function $\Psi(\mathbf{F})$ are derived from the deformation gradient $\mathbf{F}$ and physics parameters (e.g. Young's modulus $E$ and Poisson's ratio $\nu$) related to specific constitutive models. Based on Equation (1), MPM associates particles with background grids in the simulation, performing a particle-to-grid (P2G) and grid-to-particle (G2P) transfer loop. For stepping $t$ to $t + 1$, the P2G transfer can be formulated as

$$\frac{m_i}{\Delta t}(\mathbf{v}_i^{t+1} - \mathbf{v}_i^t) = -\sum_p V_p^0 \frac{\partial\Psi}{\partial\mathbf{F}}(\mathbf{F}_p^t)\mathbf{F}_p^{t\top}\nabla N_i(\mathbf{x}_p^t) \tag{2}$$

where $p$ and $i$ represent attributes for particle and grid. $V_p^0$ is the initial particle volume and $N_i(\mathbf{x}_p^t)$ is the B-spline kernel defined on $i$-th grid evaluated at $\mathbf{x}_p^t$. Grid mass $m_i^t = \sum_p N_i(\mathbf{x}_p^t)m_p$ and grid momentum $m_i^t\mathbf{v}_i^t = \sum_p N_i(\mathbf{x}_p^t)m_p(\mathbf{v}_p^t + \mathbf{C}_p^t(\mathbf{x}_i - \mathbf{x}_p^t))$ are obtained according to the standard APIC [36], where $\mathbf{C}_p^t$ is the affine matrix. The G2P transfer can be formulated as:

$$\mathbf{C}_p^{t+1} = \frac{4}{(\Delta x)^2}\sum_i N_i(\mathbf{x}_p^t)\mathbf{v}_i^{t+1}(\mathbf{x}_i - \mathbf{x}_p^t)^{\top} \qquad \mathbf{F}_p^{t+1} = (\mathbf{I} + \Delta t\sum_i \mathbf{v}_i^{t+1}\nabla N_i(\mathbf{x}_p^t)^{\top})\mathbf{F}_p^t \tag{3}$$

Afterwards, $\mathbf{v}_p$ and $\mathbf{x}_p$ are updated as $\mathbf{v}_p^{t+1} = \sum_i N_i(\mathbf{x}_p^t)\mathbf{v}_i^{t+1}$ and $\mathbf{x}_p^{t+1} = \mathbf{x}_p^t + \Delta t\mathbf{v}_p^{t+1}$.

## 4 Method

Given a monocular image, our method generates physics-grounded videos with the control signals of physics parameters and external forces. The core part of our method is a conditional diffusion model to generate physics-grounded point cloud trajectories (Section 4.1) with physics parameters and external forces as conditioning. To enable that, as illustrated in Figure 2, we first lift the input image into 3D points (Section 4.2). Once we obtain the generated trajectories, we leverage them as the condition to pre-trained video models for image-to-video synthesis (Section 4.2).

### 4.1 Physics-Grounded Generative Dynamics

Our goal is to learn the distribution of physical dynamics across various materials — termed *generative dynamics* — using a diffusion-based model, thereby avoiding the high cost, hyperparameter sensitivity,

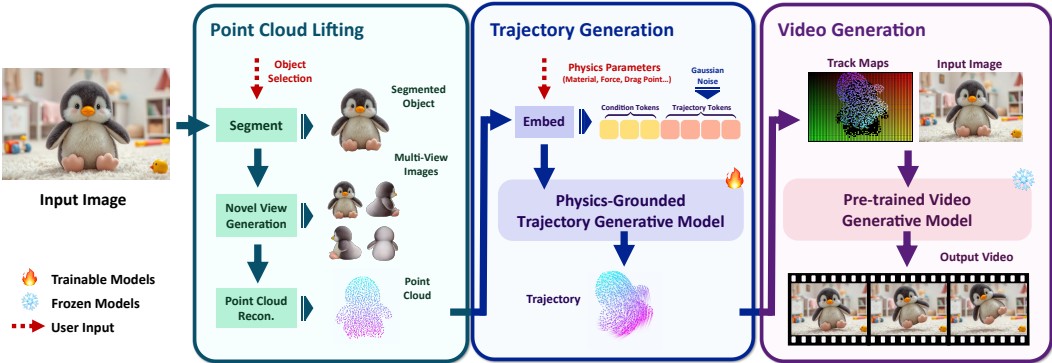

Figure 2: **An overview of PhysCtrl**. Given a single image, we first lift the object in that image into 3D points. We then generate physics-grounded motion trajectories conditioned on physics parameters and external force with a diffusion model, which are then used as strong physics-grounded guidance for image-to-video generation.

numerical instabilities, and generality–accuracy trade-offs of classical simulators. We select point clouds as our representation because they flexibly model diverse materials and suffice to control pretrained video models. Specifically, each object is represented by 2048 points in practice; we predict their trajectories over time and use them as control signals for video synthesis. We use 2048 points for guiding video model because prior work [24] show that it can achieve similar results with more points. Also, works on 4D reconstruction and generation [34, 46, 97] demonstrated that real-world motion can be represented with a sparse number of basis or control points.

### 4.1.1 Problem Setting

Given an object, represented as a 3D point cloud with $N$ points $\mathbf{P}_0 = \{\mathbf{x}_i^0 \in \mathbb{R}^3\}_{i=1}^N$, and its physics parameters $\{E, \nu\}$, our trajectory generative model generates its dynamics given an initial force. Specifically, the dynamics of the object is represented by the position of each point in future $F$ timesteps $\mathcal{P} = \mathcal{P}^{1:F} = \{\mathbf{P}^f\}_{f=1}^F = \{\{\mathbf{x}_p^f\}_{p=1}^N\}_{f=1}^F$. Denote the force, drag point and boundary condition (floor height) as $\mathbf{f} \in \mathbb{R}^3$, $\mathbf{D} \in \mathbb{R}^3$, and $h \in \mathbb{R}^1$. Thus, the goal of PhysCtrl is to predict $\mathcal{P}$ under the condition $c = \{\mathbf{P}_0, \mathbf{f}, \mathbf{D}, \{E, \nu\}, h, [\text{mat}]\}$. Here, we use an additional [mat] token to denote different materials. In this paper, we cover four different materials: elastic, plasticine, sand, and rigid. Notably, because of our flexible point cloud representation, the model is not limited to these four categories and can be readily extended to other materials, such as fluids, given sufficient computational resources.

We train our trajectory generative model on data from physics simulators—MPM [37] and a rigid-body solver. Simulator hyperparameters (e.g., substeps, grid size) introduce variability that our model, conditioning only on core physics parameters, does not capture directly. To account for this uncertainty, we employ a diffusion model to learn the conditional distribution $p(\mathcal{P}|c)$. Our method can also be extended to learning physics from more simulation methods since it requires only sampled points.

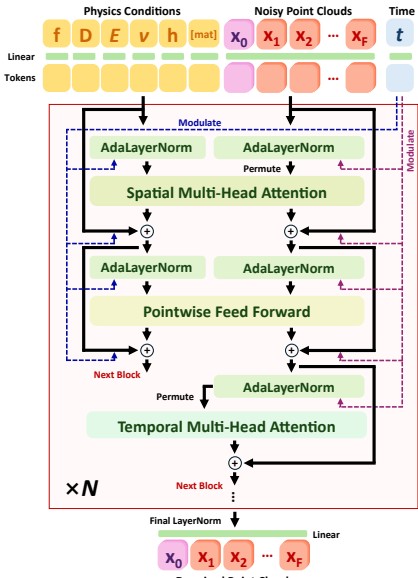

Figure 3: Our trajectory generation architecture which consists of spatial attention and temporal attention in each block.

### 4.1.2 Physics-grounded Trajectory Generative model

Prior trajectory generative models for human motion synthesis [79, 100] typically project all point positions into a single latent space, applying attention to only temporal correlations. This approach is inadequate for our setting (see Figure 1), as it overlooks crucial spatial relationships. While naive 4D attention across both space and time can model spatio-temporal correlations in physics simulation

data, it is suboptimal in terms of quality and efficiency due to the combinatorial explosion of spatial points across time steps. Instead, since we aim to model point cloud trajectories with a one-to-one point correspondence across frames, we introduce an efficient attention mechanism tailored for physics simulation data, which first applies spatial attention followed by temporal attention. This design reduces the computational complexity and, more importantly, reflects the underlying process of physics simulation: first integrating information from neighboring points, then propagating forward in time dimension.

Specifically, given noisy point cloud sequences, we apply point embedding and project it to latent dimensions, add sinusoidal positional embeddings in both space and time and predict its trajectory offset with our denoising network $\mathcal{D}$. The core of network $\mathcal{D}$ is a diffusion transformer consisting of a set of spatial-temporal attention blocks as shown in Figure 3. Each block contains two attention layers: spatial attention and temporal attention.

Spatial attention learns the correlation of each point with other points in the same frame with self-attention. To inject physical conditioning $c$ into the attention layer, we first map them into additional tokens using MLPs: $\mathbf{cond} = \text{MLP}_{\text{phys}}([\mathbf{f}; \mathbf{D}; \{E, \nu\}, h, [\text{mat}]]) \in \mathbb{R}^{d_c}$. Then, we concatenate them with point positions along the sequence dimension. Motivated by CogVideoX [94], we apply the adaptive layer norm to positional tokens and physical tokens separately to facilitate the alignment between the two spaces:

$$\hat{\mathbf{P}}^f = \text{SelfAttn}\left(\text{AdaLN}([\mathbf{P}^f; \mathbf{cond}])\right), \quad \forall f \in [1, F] \tag{4}$$

Temporal attention mainly aggregates information of the same point across all timesteps for temporal consistency. We also apply attention to the input point cloud $\mathbf{P}_0$ for better trajectory learning.

$$\hat{\mathbf{T}}_p = \text{SelfAttn}\left(\text{AdaLN}([\mathbf{T}_p])\right), \quad \forall p \in [1, N] \tag{5}$$

where $\mathbf{T}_p = [\mathbf{x}_p^0, \mathbf{x}_p^1, \mathbf{x}_p^2, \ldots, \mathbf{x}_p^F] \in \mathbb{R}^{(F+1) \times d}$.

### 4.1.3 Training Losses

We train a standard diffusion model in which we add Gaussian noise $\epsilon$ of different levels $t$ to the entire point cloud sequence: $\mathcal{P}_t = \alpha_t \mathcal{P} + \sigma_t \epsilon$ and then feed the noisy point cloud sequence into the denoising network $\mathcal{D}$. We use the signal-prediction formulation of diffusion models: $\hat{\mathcal{P}} = \mathcal{D}(\mathcal{P}_t, t, c)$.

**Diffusion Loss** We use MSE loss between the predicted and ground truth signal given noise samples:

$$\mathcal{L}_{\text{diff}} = \mathbb{E}_{\mathcal{P} \sim q(\mathcal{P}|c), t \sim [1, T]} \|\mathcal{D}(\mathcal{P}_t; t, c) - \mathcal{P}\|_2^2 \tag{6}$$

**Velocity Loss** We regulate the velocity across two frames, similar to that used in MDM [79]:

$$\mathcal{L}_{\text{vel}} = \frac{1}{F-1} \sum_{f=1}^{F-1} \|(\mathcal{P}^{f+1} - \mathcal{P}^f) - (\hat{\mathcal{P}}^{f+1} - \hat{\mathcal{P}}^f)\|_2^2 \tag{7}$$

**Physics Loss** To enable the model to learn physics-plausible motion trajectories, we introduce a physics-based supervision as regularization to enforce physical plausibility for the elastic, plasticine and sand material from MPM. Specifically, we constrain the position and velocity of the predicted points to adhere to the deformation gradient update (Equation (3)) across frames:

$$\mathcal{L}_{\text{phys}} = \frac{1}{N(F-2)} \sum_{f=1}^{F-2} \sum_{p=1}^{N} \|\mathbf{F}_p^{f+1} - g(\hat{\mathbf{x}}_p^f)\mathbf{F}_p^f\|_2 \quad g(\hat{\mathbf{x}}_p^f) = \mathbf{I} + \Delta T \sum_i \hat{\mathbf{v}}_i^{f+1} \nabla N(\mathbf{x}_i - \hat{\mathbf{x}}_p^f)^\top \tag{8}$$

where $\mathbf{F}_p^{f+1}$ and $\mathbf{F}_p^f$ are the ground-truth deformation gradient between adjacent frames and $\hat{\mathbf{x}}_p^f \in \hat{\mathcal{P}}^f$ is the predicted position. To obtain an approximation of grid velocity $\hat{\mathbf{v}}_i^{f+1}$ in Equation (8), we perform one P2G and G2P step (Equation (2)) at each frame in training. This can be formulated as

$$\hat{\mathbf{v}}_i^{f+1} = \frac{\sum_p N_i(\hat{\mathbf{x}}_p^f) m_p (\hat{\mathbf{v}}_p^{f+1} + \mathbf{C}_p^f(\mathbf{x}_i - \hat{\mathbf{x}}_p^f))}{\sum_p N_i(\hat{\mathbf{x}}_p^f) m_p} \tag{9}$$

where $\mathbf{C}_p^f$ is also from ground-truth and $\hat{\mathbf{v}}_p^{f+1} = (\hat{\mathbf{x}}_p^{f+2} - \hat{\mathbf{x}}_p^f)/(2\Delta T)$. Note that we ignore the stress term and use next-frame point velocity $\hat{\mathbf{v}}_p^{f+1}$ because it yields a more accurate approximation when the frame interval $\Delta T$ is much larger than the substep interval $\Delta t$ for MPM simulation.

**Boundary Loss** To enforce the boundary condition of the ground, we add a penetration loss, preventing the points from passing through the surface:

$$\mathcal{L}_{\text{floor}} = \frac{1}{N} \sum_{f=1}^{F} \sum_{p=1}^{N} \left( \max(h - \hat{\mathbf{x}}_p^f, 0) \right)^2 \tag{10}$$

Overall, our training loss is: $\mathcal{L} = \mathcal{L}_{\text{diff}} + \lambda_{\text{vel}}\mathcal{L}_{\text{vel}} + \lambda_{\text{phys}}\mathcal{L}_{\text{phys}} + \lambda_{\text{floor}}\mathcal{L}_{\text{floor}}$.

## 4.2 Physics-grounded Image-to-Video Generation

Starting with a single image of 3D scene with objects, we first segment out [39] the objects and generate novel view images for each object. We then feed both the novel views and the segmented image into a multiview Gaussian reconstruction model [78] and extract a point cloud for the input objects. For input with floor conditions, we support user input to select the floor region and use VGGT [83] to reconstruct the 3D scene. Then we align the coordinate system of VGGT and the 3D points of the object and obtain the height of the floor using principal component analysis. We then use our trajectory generative model to generate the dynamics of object points. The generated 3D point trajectories are then projected to the image space of the input camera viewpoint to obtain the motion trajectories of each pixel. The projected pixel trajectories can be directly used as conditioning signals for a pre-trained video generative model to produce the final video. Specifically, we use DaS [24] as the video model. It takes a "tracking video" as condition, which is the projected 3D point trajectories of 2D grid anchor points at the first frame. For each anchor point, we associate it with the nearest 3D object point. Then, we project the 3D point trajectories into 2D and get the final tracking video.

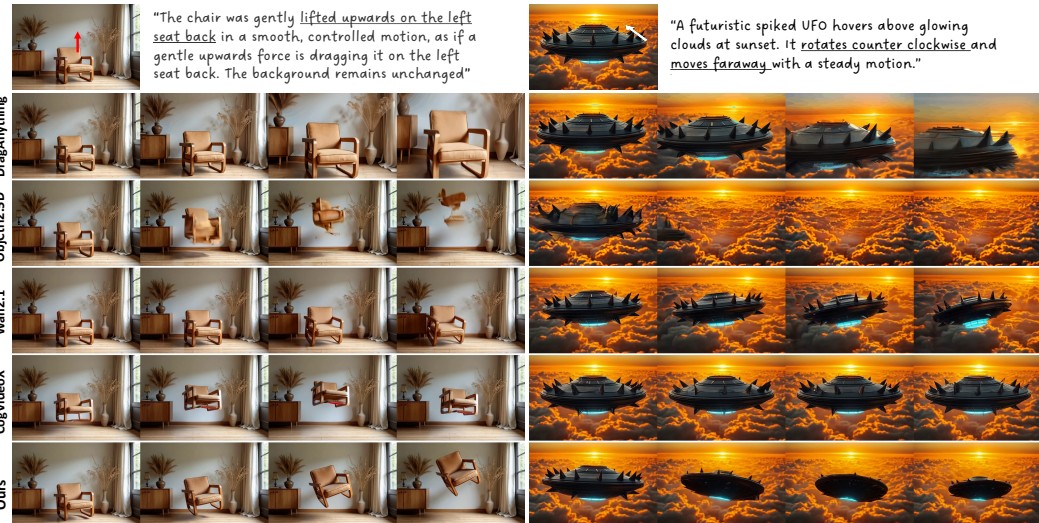

Figure 4: Qualitative comparison between our method and existing video generation methods.



Figure 5: PhysCtrl generates videos of the same object under different physics parameters and forces.

# 5 Experiments

## 5.1 Evaluation on Image-to-Video Generation

**Baselines** We compare PhysCtrl with state-of-the-art controllable video generative models, including Wan2.1-I2V-14B [82], CogVideoX [94], DragAnything [89], ObjCtrl-2.5D [87]. The first two methods support image-to-video generation with text prompts. We use ChatGPT-4o to generate text prompts based on the direction of the object movement. The last two achieve controllable video generation with user-specified single-point trajectories. We use the trajectories of the drag point generated by our model to prompt the model.

**Quantitative Evaluation** Since we are the first method to inject physics prior into a video model, we utilize GPT-4o to evaluate three aspects of 12 generated videos in a 5-Likert score inspired by VideoPhy [2]: (1) Semantic Adherence (SA): how well the content and motion in the video match the description in the text prompt, especially the alignment with the force direction and position; (2) Physical commonsense (PC): whether the object's motion follows intuitive, physically plausible dynamics given the applied force direction and position; (3) Video Quality (VQ): overall visual and temporal quality of the video. Results in Table 1 show that our method achieve the best results across all baselines. Results of user study can be found in the supplemental.

**Qualitative Evaluation** The qualitative results between our method and baselines can be found in Figure 4. CogVideoX-5B [94] and Wan2.1 [82] have strong generation ability and partly follow the text prompts. However, they only use text prompts as conditions and lack precise control, thus, they cannot produce motions that fully reflect physics conditions. For example, the *chair* in Figure 4 doesn't move according to the force direction. DragAnything [89] uses purely 2D trajectories and cannot distinguish between camera motion and object motion, thus sometimes generating camera motions while objects remain static. More importantly, both DragAnything [89] and ObjCtrl2.5D [87] only use coarse trajectory as a condition and struggle to generate more complex motions, *e.g*, the UFO case in Figure 4 that contains both rotations and depth change. In comparison, PhysCtrl produces physics-plausible videos that follow the given forces by generating physics-grounded 3D trajectories as a strong conditional signal to guide the superior generation capability of pretrained video generative models for video synthesis.

Table 1: Results of video evaluation.

|  | SA↑ | PC↑ | VQ↑ |
|---|---|---|---|
| DragAnything [89] | 2.9 | 2.8 | 2.8 |
| ObjCtrl [87] | 1.5 | 1.3 | 1.4 |
| Wan2.1 [82] | 3.8 | 3.7 | 3.6 |
| CogVideoX [94] | 3.2 | 3.2 | 3.1 |
| Ours | **4.5** | **4.5** | **4.3** |

Table 2: Quantitative comparison on trajectory generation.

| Method | vIoU↑ | CD↓ | Corr↓ |
|---|---|---|---|
| M2V [6] | 24.92% | 0.2160 | 0.1064 |
| MDM [79] | 53.78% | 0.0159 | 0.0240 |
| Ours | **77.59%** | **0.0028** | **0.0015** |

**Results on Varying Physical Conditions** Since our trajectory generative model is conditioned on external forces and physics parameters, we can generate videos of the same object under varying conditions. As shown in Figure 5, we can change the Young's modulus in elastic material to produce results with different deformations given the same force. The direction and amplitude of the force can also be adjusted to match the user's desired motion. We found that Poisson's ratio $\nu$ has negligible influence on the generated trajectories, similar to the findings in PhysDreamer [97].

## 5.2 Evaluation on Generative Dynamics

**Baselines** We compare our approach with existing methods that focus on generative dynamics, including Motion2VecSets [6] and MDM [79]. Motion2VecSets is a method for reconstructing sparse point cloud sequences; we eliminate the sparse point cloud condition and introduce physics conditions instead. MDM is primarily aimed at human motion generation, so we substitute human joints with point clouds and incorporate physics conditions as additional tokens. For computation efficiency, we trained all baselines and ablations on our elastic subset of 160K objects that contains complex deformations for metrics comparison.

**Evaluation Metrics** Following [45, 6], we adopt volume Intersection over Union (vIoU), Chamfer Distance (CD) and $L_2$-distance error for evaluation. vIoU measures the overlap between predicted and ground truth point clouds, CD measures the averaged per-point pairwise nearest neighbor distance

between two point clouds, $L_2$-distance is the Euclidean distance between two corresponding point clouds. Each metric is calculated at each timestep separately and averaged across all frames.

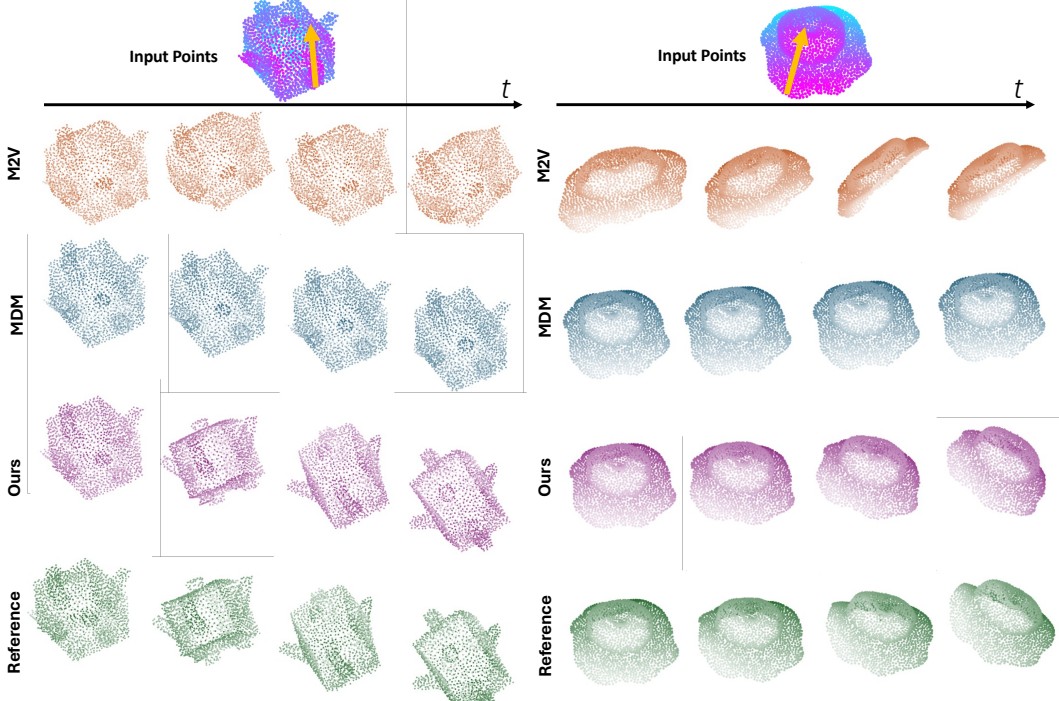

Figure 6: **Qualitative results**: Compared to baselines, our method enables high-quality and coherent generation of motion sequences from physics conditions and closely matches the reference.

Table 3: Ablation study on trajectory generative model.

| Method | vIoU↑ | CD↓ | Corr↓ |
|---|---|---|---|
| w/o spatial attention | 33.76% | 0.2348 | 0.1163 |
| w/o temporal attention | 53.63% | 0.0480 | 0.0507 |
| w/o physics loss | 76.30% | 0.0030 | 0.0016 |
| Ours | **77.59%** | **0.0028** | **0.0015** |

Table 4: Ablation study on using traditional simulator and our model for video generation.

| Method | SA↑ | PC↑ | VQ↑ |
|---|---|---|---|
| Traditional (2048 points) | 4.3 | 4.3 | **4.4** |
| Traditional (8192 points) | 4.3 | 4.3 | 4.2 |
| Ours (2048 points) | **4.5** | **4.5** | 4.3 |

**Results** Table 2 shows the quantitative comparison of our method with other baselines. Our method demonstrates the best performance over all metrics on the testing set. The qualitative comparison can be found in Figure 1. Our model achieves physics-grounded and consistent generation of motion trajectories. Motion2vecsets struggles to generate time-coherent motions because in our experiments, there is no sparse point cloud condition in their original setting. M2V struggles to generate coherent motions in our experiments. There are two potential reasons for this. Firstly, their model is originally designed for point cloud completion, but in our setting, there is no sparse point cloud condition. Prior work [98] also found that M2V does not work well in this situation. Secondly, their deformation latent is encoded frame-by-frame without temporal interaction. MDM can generate consistent motion sequences, but fails to capture detailed deformations because all points in a frame are projected into a single latent. The superiority of our method is based on our spatial-temporal attention block, which leverages explicit per-point correspondence.

### 5.3 Ablation Study

The qualitative and quantitative results of the ablation study for trajectory generation can be found in Figure 7 and Table 3. Our physics loss improved all the metrics and makes the results of our trajectory generation close to the ground truth. The physics loss aligns the updated deformation gradient with the ground truth and constrains the predicted positions. Although without physics loss, our model can achieve good results, it can be further improved with physical guidance as regularization.

Table 4 presents the ablation study for video generation. Results show that using our trajectory generation model for video generation is on par with using a traditional simulator. Also, results also show that using more points didn't bring a performance gain for video generation.

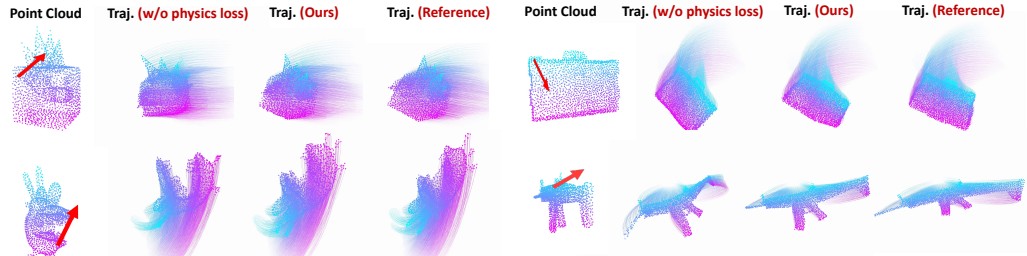

Figure 7: Comparison of using physics loss on trajectory generation. Here we show the final point position and tracks for points. With physics loss, the results are more closely aligned with the reference (simulated by MPM).

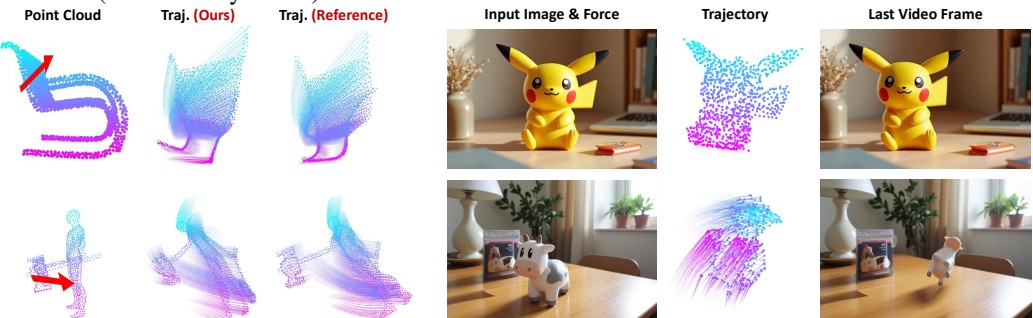

Figure 8: Failure cases.

## 6 Discussion

**Failure Cases** As shown in Figure 2, our model contains three components: point cloud lifting, trajectory generation and video generation. (1) **Failure due to point cloud lifting is extremely rare**. Single image-to-3D produces reasonable geometry overall and is unlikely to yield severely distorted or implausible shapes. While minor artifacts (e.g. geometry not perfectly smooth or noisy points on the surface) may occur, they have minimal impact on our results. We achieve such robustness to geometric variations because our trajectory generation model on the diverse Objaverse dataset and applying data augmentation of surface noise. (2) **Failure cases of our trajectory generation model**: our model cannot handle thin structures well and sometimes might fail to accurately capture complex internal deformations. (3) **Failure cases due to video generation**: The video model cannot fully follow the trajectory when it conflicts with the prior ofthe video model. For example, when the user input for the object material is very stiff, but it appears soft according to RGB information. The video model might also hallucinate unexpected content in the occluded region, *e.g*, for an animal, it might generate five legs. See Figure 8 for example failure cases.

**Extension to Multiple Objects** MPM achieves multi-object interaction by representing scene dynamics as point movement. Our model also predicts point trajectories, so it is also inherently capable of multi-object interaction. We did a preliminary experiment on multi-objects on a simplied setting: we create a dataset that has an object dragged towards a cube and colliding with the cube from different angles and distances. We trained our model on it and achieved 93.70% vIOU on the held out testing set.

## 7 Conclusion and Limitations

In this paper, we introduce PhysCtrl, a novel framework for physics-grounded video generation with physics parameters and force control. We design a diffusion model with spatial-temporal attention blocks and physics-based supervision to effectively and efficiently learn complex physical deformations directly on point cloud sequences. The generated motion trajectories can be used as a strong conditional signal for pre-trained video generative models. Our experiments demonstrate that PhysCtrl can generate physics-grounded dynamics and enable high-quality image-to-video generation results conditioned on external forces and physics parameters.

Our approach mostly focuses on single-object dynamics for four material types and does not cover all possible materials. We only do initial study on multiple objects and more complex phenomena should be investigated, such as intricate boundary conditions. Future work includes addressing these limitations and extending PhysCtrl to more diverse and complex physics phenomena in the real world.

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

The supplementary material covers the following sections: Implementation Details Appendix A, User study Appendix B, Physics Parameter Estimation Appendix C, Results Appendix D, Societal Impacts Appendix E, Data and Model Safeguards Appendix F. **We also encourage readers to refer to our supplementary videos for demonstrations of animatable results.**.

# A  Implementation Details

**Dataset.**  To make our model handle diverse objects and motion trajectories, we generate data using physics simulation using high-quality 3D objects selected from ObjaverseXL [16, 15]. We simulate animations for each object with the MPM simulator [37] as the ground-truth. We use a fixed number of simulated points $N = 2048$ (uniformly sampled on the faces of the mesh) and frames $F = 24$ to align with our model's input. For data augmentation, we randomly rotate the object around $y$-axis and add noise $\epsilon_p^{aug} \sim \mathcal{N}(0, 0.01^2)$ to each sampled initial point. Our whole dataset contains 550K objects, including 150K elastic objects of different drag force directions, 100K objects of gravity across elastic, sand, plasticine and rigid respectively. For the simulated animation of varying drag force, we randomly sample a constant force $\mathbf{f}$, a drag point $\mathbf{D} \in \mathbf{P}_0$ and physics parameters $E \in [10^4, 10^7]$, $\nu \in [0.05, 0.45]$. The force $\mathbf{f}$ has an outward direction of the object surface and a magnitude between $0.02G$ and $0.3G$ in total ($G$ is the gravity of the whole object) and is only applied to points close to the drag point $\mathbf{D}$.

**Training**  For metric comparison and ablation, we train our base model on the 150K elastic subset that contains different force and physical parameters with 6 layers and 256 latent size on 8 NVIDIA L40 GPUs with 48GB GPU memory for 60K iterations with a total batch size of 32, which takes about 30 hours. We randomly leave out 100 animations from this dataset as the test set and keep the remaining ones for training. We train a large model of different materials with 12 layers and a 512 latent size on all 550K data with the same iterations and batch size, which takes about 80 hours. We use AdamW optimizer with betas (0.9, 0.999) and a learning rate of 1e-4 with a cosine schedule and a warmup of 100 steps. We clip the gradient with the maximum norm of 1.0 and train with bfloat16 precision. We use a DDIM scheduler for sampling. For 25 diffusion steps, it takes 1s and 3s for the base and large model. For 4 diffusion steps, it takes 0.13s and 0.48s for the base and large model. We found that 4 steps can already achieve great results due to the low uncertainty of the model.

**Image-to-3D Pipeline**  We use SAM [39] to segment the object in the input image and run SV3D [81] to generate 20 novel-view images of that object with orbit camera poses, from which we pick three images with azimuth $(90°, 180°, 270°)$ relative to the input and send them together with the input into LGM [78] for 3D Gaussian reconstruction. We then convert the 3D Gaussians to a plain point cloud and sample $N$ points using farthest point sampling (FPS) for trajectory generation.

**GPT-4o Evaluation**  We prompt GPT-4o with the following prompt to use it for evaluation (Results might vary with GPT updates):

```
You are tasked with evaluating the quality of image-to-video generation
produced by a model.
For each test case, you will be given: 1. A text prompt describing
a single object and a force applied to it.  The force's position
and direction are visualized as a red arrow in the input image.  2.
An input image of the object.  3. Five sets of 10 evenly spaced
frames-each set corresponds to a video generated by a different model
from the same input.
Please evaluate this video based on the following three criteria using a
5-point Likert scale (1 = poor, 5 = excellent):
- Semantic Adherence:  How well the content and motion in the video
match the description in the text prompt, especially the alignment with
the force direction and position.  Note that the video should starts
with the input image.
- Physical Commonsense:  Whether the object's motion follows intuitive,
physically plausible dynamics given the applied force direction and
position.
```

Table 5: Results of user study.

| | Ours | CogVideoX | Wan | DragAnything | ObjCtrl2.5D |
|---|---|---|---|---|---|
| Physics Plausibility | **81.0%** | 5.5% | 10.2% | 1.2% | 2.1% |
| Video Quality | **66.0%** | 6.2% | 18.3% | 4.5% | 5.0% |

```
- Video Quality:  The overall visual and temporal quality of the video
(note that static or nearly-static sequences are less preferred).
Provide your evaluation for each video strictly in the following
one-line format:
Video i, Semantic Adherence score, Physical Commonsense score, Video
Quality score
```

## B    User Study

We conducted a user study to evaluate the physics plausibility and overall quality of the videos generated by our model and other baselines. The study consisted of 12 questions, each including an input image with the force location and direction marked on the image, a text prompt describing the image and applied force, and generated video results produced by five different methods. The users are asked to carefully observe the videos and evaluate them from two aspects: (1) **Physics plausibility**: select the one that best matches the force direction (red arrow) and corresponding text prompt. The force and text prompt are assumed to match each other. (2) **Overall Video quality**: Select the one that has the best visual and temporal quality.

We received a total of 35 responses (35 × 12) and computed the percentage of times each method was selected as the best-performing video for each question. The results are summarized in Table 5, showing the preference rates for each method. The findings indicate that our model consistently outperforms baseline methods in terms of both physics plausibility and video quality. Although Wan received the second-best video quality, some of these high-quality videos suffer from low physics plausibility.

## C    Physics Parameter Estimation

Our trained trajectory generation model learns the conditional distribution of physically plausible motion trajectories, so it can also be used for inverse problems, *i.e*, to estimate the condition $c$ given ground truth trajectories $\mathcal{P}$. The intuition is that **a $c$ that is closer to the ground truth will introduce less discrepancy between the denoised trajectories and ground truth trajectories**. To this end, we define an energy function that measures how well the model can denoise a noisy version of $\mathcal{P}_t$ under that condition:

$$\mathcal{E}(c) = \mathbb{E}_{t \sim [1,T]} \|\mathcal{P}_t - \mathcal{D}(\mathcal{P}_t; t, c)\|^2, \tag{11}$$

During optimization, the denoiser $\mathcal{D}$ is frozen and only $c$ is optimizable. We add random noise to the ground truth trajectory and feed it into the trained network to denoise. The gradient of the energy function will be backpropagated to optimize $c$.

We simulate 15 trajectories for elastic materials to test our physics parameter estimation pipeline. We compare our method with differentiable MPM [32], which needs to accumulate gradients over hundreds of substeps for one backward pass (costing more than 3min compared to 0.1s for ours). Table 6 shows that our method only takes about 2 minutes while achieving relatively good results, which also **demonstrates that our trained diffusion model captures physics-plausible motion trajectories**.

## D    More Results

More results of our method and baseline comparisons can be found in Figure 9. **We strongly encourage the readers to look at our video for better comparison, as isolated frames cannot fully represent the physical dynamics well.**

Table 6: Mean Absolute Error (MAE) of Young's Modulus on physics parameter estimation.

| Method | Runtime (min.) | MAE of $\log_{10}(E)$ |
|---|---|---|
| Ours | 2 | 0.506 |
| Diff. MPM (5 iters) | 20 | 0.439 |
| Diff. MPM (15 iters) | 60 | 0.394 |

## E  Societal Impacts

**Positive Impacts** Our method integrates physically grounded simulation signals into video generative models, offering new avenues for controllable and physically plausible video synthesis. These can support people from amateurs to filmmakers and designers in rapidly prototyping ideas with accurate physical behavior, democratizing access to high-fidelity visual tools.

**Negative Impacts** High-fidelity generative models, especially when conditioned on physical signals, may be misused for creating deceptive content such as realistic yet fabricated disaster footage or physically plausible fake videos. This poses risks for misinformation and erosion of public trust. Although our approach enhances physical plausibility, it is important to note that the generated outputs are not real-world occurrences.

## F  Data and Model Safeguards

Given the dual-use nature of video generation models, we recognize that our pretrained model could be misused to generate deceptive, physically plausible videos for misinformation. As such, we will implement appropriate safeguards to support controlled access when we release our model, including: (1) requiring users to agree to usage guidelines and restrictions, (2) distributing the model under a research-only license, (3) investigating automatic safety filters that can flag potentially harmful uses. These steps aim to reduce the risk of malicious or unintended applications while still supporting reproducible research.

Our training data consists exclusively of synthetic point cloud trajectories representing object motion under simulated physics. These datasets contain no images, videos, or human-related content, and thus should pose no risk of visual misinformation, privacy violations, or unsafe content. All point clouds are generated in simulation environments and contain only geometric and physical information about object movement.

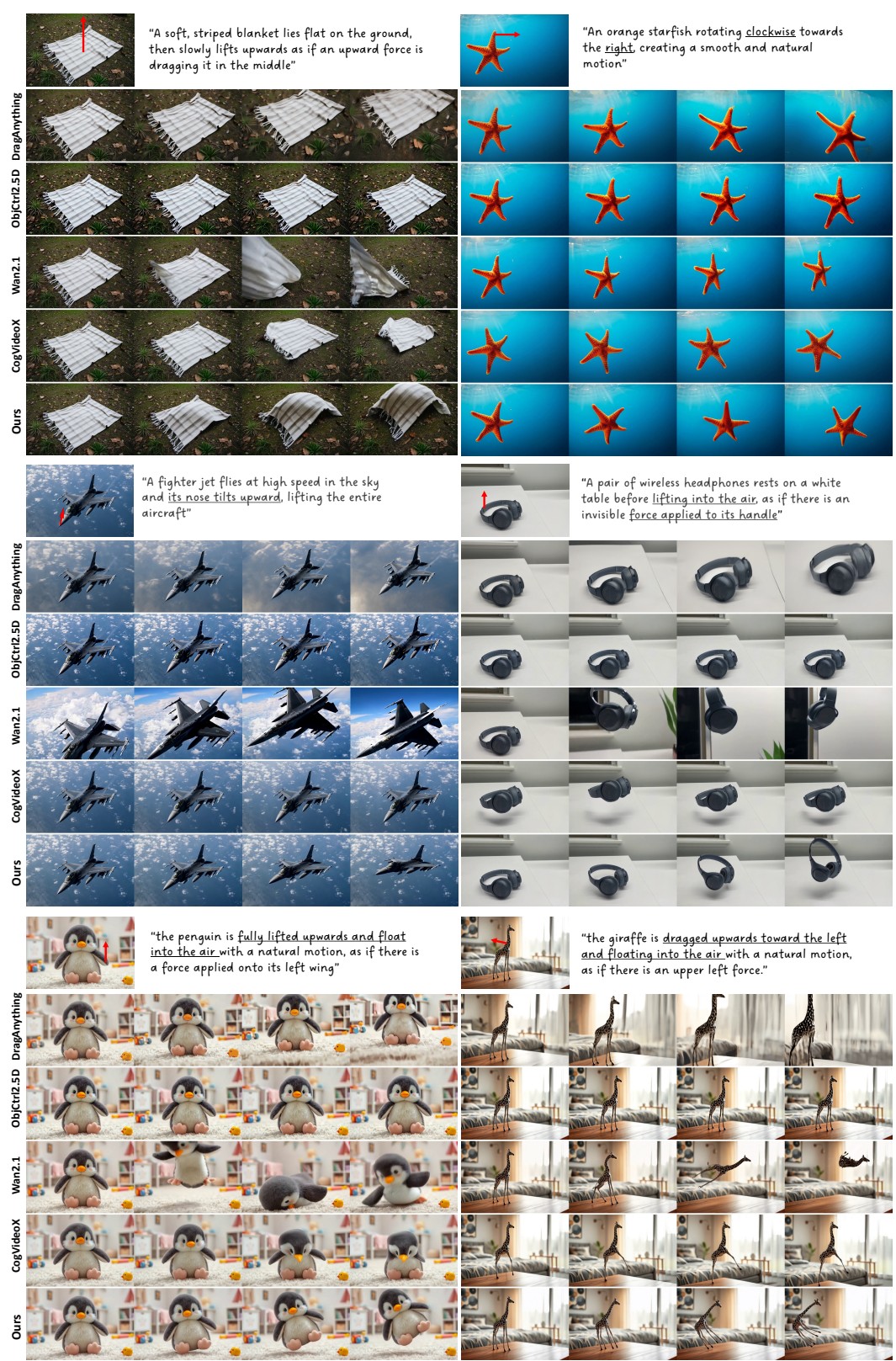

Figure 9: More qualitative comparison between our method and baselines.

