# OpenReview forum: "PhysCtrl: Generative Physics for Controllable and Physics-Grounded Video Generation"
_NeurIPS.cc/2025/Conference — NeurIPS 2025 poster_

### Official Review · Reviewer_eLwH · 2025-06-25

**Clarity:** 3
**Significance:** 3
**Originality:** 3
**Rating:** 5
**Confidence:** 4

**Summary:**

This paper proposes a way to do physics-controllable i2v generation. As I understand it, the main contribution of this paper is a novel way of generating point trajectories for objects in response to user-provided physics parameters and applied forces. They propose a diffusion model (which they call a "generative physics network") which inputs physics conditions and a point cloud, and outputs the position of each point in future timesteps. I.e. they propose an auxiliary model for "approximate" physics simulation, as a representation bottleneck. They train this model using a synthetic dataset of size 550k generated by physics simulators. Finally, they feed this point cloud into some else's pretrained video diffusion model (Diffusion as Shader) which accepts pixel trajectories as conditioning. The end-to-end result is a (force+image+text)2video generation model.

**Questions:**

1. Can you please explain exactly how you used the Diffusion as Shader baseline?
2. The description in lines 49-59 of the diffusion model ("... it first aggregates spatial influences from neighboring points...") sounds very related to graph neural networks, did you explore that connection at all?
3. It seems to me like the physics loss is the most complicated part of the paper (there's half a page of math there) and it seems the least important to the downstream performance? In Table 3, the row without physics loss is barely worse than the other rows, whereas the spatial attention and temporal attention are clearly crucial. Is this true? If this is the case, I would strongly recommend moving the Material Point Method section in the paper to the appendix, so the reader can black box the loss function, and move brief summaries of the experiment details from the appendix (e.g. construction of synthetic dataset, design of human study) into the body of the paper.
4. What were some failure cases of the model? And were the failure cases more because an initial poor 3D point cloud was extracted? or because the diffusion model wasn't able to model it well? or because, given point cloud trajectories, the Diffusion as Shader wasn't able to accept them as conditioning in a good way?
5. More generally, were certain classes of demos more or less likely to succeed and fail? E.g. Diffusion as Shader is built on top of CogVideoX... did the video prior make certain classes of demos more or less likely to succeed? Or more generally, what is the significance of the choice of the video prior for the base video model?
6. In line 262, should the sentence read: "... cannot produce motions that fully reflect *physical conditioning*"? Or am I misunderstanding the claim you're trying to make in that sentence?

**Ethical Concerns:**

["NO or VERY MINOR ethics concerns only"]

**Final Justification:**

I already had a high opinion of the paper before the rebuttal period, due to the novelty + utility of the task (force-conditioned video generation), and the high quality of the output videos. I also thought that the experiment design was reasonable and strong. My questions to the authors were mainly about experiment design, and those have all been addressed.

**Limitations:**

(Yes)

**Paper Formatting Concerns:**

(No concerns)

**Quality:**

4

**Strengths And Weaknesses:**

Overall, I think this paper is excellent, I intend to advocate strongly for this paper to be accepted.

Strengths:
1. This paper approaches a very timely modeling problem (physics-conditioned video generation) with a novel and interesting technique that, notably, doesn't involve using any physics simulator at inference (although it does make use of 3D priors at inference, as a representational bottleneck). It makes good use of building on top of prior works while also proposing a new architecture (the force-conditioned point cloud diffusion model) which can be used modularly, for many different downstream tasks.
2. I think that the choice of baselines is very effective. Two text-based (Wan 2.1, CogVideoX), and two single-point trajectory-conditioned (DragAnything, ObjCtlr-2.5D). Comparing to them clearly makes the point that PhysCtrl is superior.
3. Likewise, the evaluation metrics for the diffusion model are very clear.
4. I thought the writing and motivations were very clear, and the paper was a pleasure to read. I especially thought that the explanation of the Problem Setting (4.1.1) was explained very clearly. More generally, I think the organization of the paper is excellent (modulo a few potential suggestions for improvement, see below)

Weaknesses:
1. I thought some experimental details were swept under the rug, which made some of the takeaways unclear (see Questions section).
2. Some of the phrasing was a tad vague and misleading. E.g. lines 60-61: "we further show that the generated trajectories can drive pretrained video models for synthesizing physically plausible i2v generation". By the time I got to the end of the paper I understand what was meant by "drive" (namely, plugging in the generated trajectories into someone else's trajectory-conditioned model) but I think this should be made more specific up front, or else it's not clear what is this paper's specific contribution. Additionally, I think that the claim on line 251 ("we are the first method to inject physics prior into a video model") is false/misleading; there are plenty of works that do physics-based video generation, including papers cited by the present work (e.g. Physdreamer). Is there perhaps a different way to word that phrase that more precisely describes the novelty proposed by this paper?
3. (Minor weakness, more subjective) it would be nice if in the beginning of the Method section, when it discusses how the method works, and how the subsequent subsections correspond, for those to be in agreement/chronological order during training/inference (i.e. not going from 4.2 to 4.1 to 4.2. This just made it a tad harder for me to follow the logic on the first read)

---

> ### Author Rebuttal · Authors · 2025-07-31
>
> Thanks for your insightful comments for improving our work. We appreciate your positive feedback and your constructive suggestions, which has been a source of great encouragement for us. The following are detailed responses to your concerns.
>
> > Some of the phrasing was a tad vague and misleading.
>
> We will make it specific in the beginning that we use the generated trajectories as input of a trajectory-conditioned model.
>
> > The claim on line 251 ("we are the first method to inject physics prior into a video model") is false/misleading.
>
> Our model is conceptually different from PhysDreamer from a high level. **PhysDreamer exploits physics knowledge from pretrained video models**. It estimates physical properties of a trained 3DGS using the motion priors from video models with differentiable MPM. Differently, **our model is injecting physical priors into video models** by conditioning it on our generated motion trajectories. Moreover, we show in the supplemental that our model is also able to estimate physical properties given known motion trajectories. There are concurrent works, such as WonderPlay [1], Force prompting [2] (both appear after our submission), that attempt to improve the physics plausibility of video models. We will discuss these works and adjust our tones in the revision.
>
> [1] WonderPlay: Dynamic 3D Scene Generation from a Single Image and Actions, Arxiv, May 2025
> [2] Force Prompting: Video Generation Models Can Learn and Generalize Physics-based Control Signals, Arxiv, May 2025
>
> > Beginning of Method Subsection
>
> Thanks for your advice. We will adjust the beginning of the method section to a more logical order.
>
> > How to use Diffusion as Shader.
>
> Diffusion as Shader (DaS) takes a **tracking video** as the input of the conditioning branch of ControlNet. Each frame of the tracking video is the projected 3D point trajectories of **2D anchor points (fixed grid positions for all videos)** from the first frame. Our model generates the 3D point trajectories of the object, so we need to convert it to the tracking video of DaS for inference. Specifically, for each 2D anchor point in the first frame, we associate it with the nearest 3D point in the object. Then, we can project the 3D point trajectories into 2D and generate the tracking video frame by frame. We will add these details to the next version.
>
> > Explore GNN.
>
> We chose a Transformer-based architecture due to its strong representational capacity and general effectiveness across a range of tasks. While we did not exactly test a GNN, we experimented with a variant of the Transformer that has a similar spirit of GNN, that is, each point only attends to its k-nearest neighboring points to model local relationship, which is similar to the local aggregation behavior of GNNs. We found that this approach doesn’t work well, particularly has severe error accumulation. This supports our decision to apply attention across all points, which may allow the model to better capture long-range dependencies and complex interactions.
>
> > It seems to me like the physics loss is the most complicated part of the paper (there's half a page of math there) and it seems the least important to the downstream performance? In Table 3, the row without physics loss is barely worse than the other rows, whereas the spatial attention and temporal attention are clearly crucial. Is this true? If this is the case, I would strongly recommend moving the Material Point Method section in the paper to the appendix, so the reader can black box the loss function, and move brief summaries of the experiment details from the appendix (e.g. construction of synthetic dataset, design of human study) into the body of the paper.
>
> The physical loss helps the model to generate more physics plausible trajectories. In Figure 7, we show four examples that using physical loss helps generate more physical plausible trajectories. For these cases, if there is no physics loss, generating videos with these trajectories will not look consistent with the input force. We appreciate your suggestion and will move some experiment details into the main paper.
>
> > Discussion of failure cases due to poor 3D point cloud extraction / our diffusion model / Diffusion as Shader
>
> Classification of failure cases: (1) **Failure due to initial poor image-to-3D reconstruction is very rare**. Single image-to-3D produces reasonable geometry overall and is unlikely to yield severely distorted or implausible shapes (note that we do not need appearance information). While minor artifacts (e.g. geometry not perfectly smooth or noise points on the surface) may occur, they have minimal impact on our results. We achieve such robustness to geometric variations because we trained the trajectory generation network on the diverse Objaverse dataset and applied surface noise augmentation during data generation. Moreover, the video model only needs point trajectories as motion signals and will generate the details. (2) **Failure cases of our diffusion model**: we compared our results with MPM simulation on the Objaverse testing set. We found that cases with low vIoU typically involve thin structures or scenarios where our model predicts global motion but fails to accurately capture internal deformations. (3) **Failure cases due to video model**: we found in some cases, especially when the trajectories might conflict with the prior of video model, DaS cannot fully follow our trajectories. For example, when the user input for the object material is very stiff, but it appears soft according to RGB information. Then the resulting video might still be stiff. The video model might also hallucinate some unexpected content in the occluded region. For example, for an animal, it might generate five legs.
>
> We will add these discussions and visualization of failure cases in the revision.
>
> > Did the video prior make certain classes of demos more or less likely to succeed? Or more generally, what is the significance of the choice of the video prior for the base video model?
>
> As discussed above, when the trajectories conflict with the prior of the video model, DaS might not fully follow our trajectories. We selected CogVideoX, as it was the strongest available model at the time of our submission. We think that using more powerful video models (especially those that have better world knowledge) will generally bring better results, e.g. generate more plausible details for occluded parts.
>
> > Line 264.
>
> Thanks for pointing this out. We agree that using “physical conditioning” is more accurate.

---

> > ### Comment · Reviewer_eLwH · 2025-08-04
> >
> > Thank you for your detailed reply. A few follow up questions:
> >
> > 1A. Can you please explain how to parse Figure 7? I'm having trouble understanding how it illustrates that the physics loss makes it that much better.
> >
> > 1B. Additionally, I don't understand why the introduction of the physics loss barely improves the quantitative metrics in Table 3, yet is (qualitatively) completely responsible for making the learned trajectory closer to the ground truth trajectory in Figure 7. Can you please explain how you designed the experiments which yielded the results in Figure 7 and Table 3 (i.e. did those all use both spatial attention and temporal attention, but not the physical loss?) **Are the third row of Table 3, and the columns labeled "Traj. (w/o physics loss)" in Figure 7 from the same experiment?**
> >
> > 2. Is it mentioned in the paper that you applied surface noise augmentation during data generation? (If so, where?)
> >
> > You mentioned that you plan to add failure case analysis to the next draft of the paper. I just wanted to add my 2 cents, that I think that's a great idea. I find that understanding when/where models like these fail is equally as important as understanding when/where they excel.

---

> > > ### Author Response · Authors · 2025-08-04
> > >
> > > Thank you for your feedback. Here are our answers to your follow-up questions.
> > >
> > > > Question 1A: Parse Figure 7
> > >
> > > In Figure 7, we have four examples. In each example, the first column is the **input point cloud** and **3D force directions**, the second and third columns are trajectories (visualized as final point positions with tracks attached to each point denoting one point trajectory)  without and with physics loss, the last column is trajectories of MPM. We can see that given the same input points and 3D force, using physics loss (second column) is closer to MPM in terms of final point positions and track shape, compared with without physics loss. We will add more detailed descriptions of Figure 7 of better understanding when updating our paper.
> > >
> > > > Question 1B: Explanation of physics loss and results in Figure 7 / Table 3
> > >
> > > The results in Figure 7 and Table 3 use both spatial and temporal attention, but without physics loss. Therefore, the results of w/o physics loss in Table 3 are the same experiment as in Figure 7.
> > >
> > > In Table 3, **w/o spatial attention and w/o temporal attention are related to the architecture design of our model, while physics loss is more like a regularization loss term for conservation of mass and momentum (not directly related to point-wise distance used in the three metrics)**. Our testing set is randomly sampled with different force directions, force magnitudes and materials. Some of them might be easy with nearly translation motion, while others can be hard with rotations or deformations. Since each point strongly relies on both spatial and temporal information of other points, lacking either of them would result in a huge performance drop and the model cannot work well even in simple cases. If we have both spatial and temporal attention, but without physics loss, the model can achieve relatively good results in many cases, but might still not be good in certain cases shown in Figure 7. Therefore, spatial-temporal attention with L2 pointwise distance (diffusion loss) is the building block of the model, while physics loss regularizes the model to be one step closer to the ground truth. However, according to our analysis in failure cases, for thin structures or complex internal deformations, physics loss is still not enough, better losses or architectures might be introduced in future works.
> > >
> > > > Question 2: Is it mentioned in the paper that you applied surface noise augmentation during data generation? (If so, where?)
> > >
> > > Yes. We described that in the supplementary PDF Line 10 and Line 11.
> > >
> > > > Failure cases analysis.
> > >
> > > We strongly agree that failure case analysis would be very important for understanding the method.

---

> > > > ### Comment · Reviewer_eLwH · 2025-08-05
> > > >
> > > > Thank you for your detailed responses, my questions have been addressed. I still plan to advocate strongly for this paper to be accepted.

---

### Official Review · Reviewer_F6p4 · 2025-06-29

**Clarity:** 3
**Significance:** 2
**Originality:** 2
**Rating:** 4
**Confidence:** 4

**Summary:**

This paper presents PhysCtrl, a framework for generating physics-grounded videos from a single image with explicit control over material properties and external forces. The pipeline first segments the target object, reconstructs its 3D point cloud using multi-view synthesis, and then predicts its motion using a learned trajectory generation model trained on simulated dynamics from the Material Point Method (MPM). These trajectories are projected and used to guide a pretrained video diffusion model to generate physically plausible videos

**Questions:**

As similar pipelines have been proposed in prior work, where the object is first lifted to 3D, then simulated, and finally rendered, the overall novelty of the proposed framework may be limited. Furthermore, the rendering module directly adopts an existing video diffusion model without specific adaptations or modifications. Therefore, the main contribution appears to lie in the trajectory generation module.

However, I would like the authors to clarify the necessity of training a neural trajectory predictor instead of directly using MPM, which should yield similar results for these relatively simple object animation cases. As mentioned above, users still need to manually specify inputs such as boundary conditions and external forces, which raises questions about the practical advantages of the proposed model over traditional simulation.

Additionally, since the synthetic dataset seems to include only single-object dynamics, it is unclear whether the method would generalize to multi-object interactions with collisions.

While I understand it’s not possible to compare against all baseline methods, it would be nice to include a comparison with this SOTA method [1], which also addresses motion-controllable video generation.

[1] Burgert R, Xu Y, Xian W, et al. Go-with-the-flow: Motion-controllable video diffusion models using real-time warped noise. CVPR 2025

**Ethical Concerns:**

["NO or VERY MINOR ethics concerns only"]

**Final Justification:**

After reading the rebuttal, I am raising my score to 4. I acknowledge the advantages of the neural simulator over traditional simulators in single-object dynamics, particularly in terms of speed and robustness.

**Limitations:**

yes

**Quality:**

3

**Strengths And Weaknesses:**

Strength:

1. The paper is well written and easy to follow
2. The proposed pipeline is technically sound and comprehensive
3. Expreiments are thorough and a demo video is provided to demonstrate the usage of of the proposed method.
4. The results of the proposed method look impressive, showing significant improvement over the baseline methods

Weakness:

1. The necessity of learning a trajectory generation model is not fully justified. Since the model takes inputs similar to MPM (e.g., material properties and external forces), it's unclear what advantage it offers over directly using simulation, especially for small-scale scenes where MPM is efficient
2. It is unclear how the model accounts for external elements like the ground plane if only the foreground object is segmented. For example, in Figure 1, how does the model determine that the penguin is supposed to collide with the table while the plane should stay in the air when rotating?
3. The approach seems limited to single-object dynamics. No multi-object interactions or collisions are demonstrated, and it appears the dataset does not include such scenarios.

Minor:

Some recent relevant papers could also be cited for completeness:

[1] Burgert R, Xu Y, Xian W, et al. Go-with-the-flow: Motion-controllable video diffusion models using real-time warped noise. CVPR 2025

[2] Xie T, Zhao Y, Jiang Y, et al. PhysAnimator: Physics-Guided Generative Cartoon Animation. CVPR 2025

---

> ### Author Rebuttal · Authors · 2025-07-31
>
> Thank you for your insightful comments and questions, which is crucial to improving the quality of our work. The following are detailed responses to your concerns.
>
> > The model takes inputs similar to MPM and the necessity of training a neural trajectory predictor is not clear.
>
> We would like to first clarify that **our user inputs are less than MPM**. We only require minimal information: force and material information, while MPM requires many other hyperparameters: grid size, damping scale for grid velocity, frame dt and substep dt etc. Besides this, using our model has several other benefits: (1) **More robust**: we use a traditional simulator to generate training data by tuning their parameters and filtering failure cases. The filtered training data helps us train a robust neural simulator, even for cases where classical simulators struggle. For example, we found that when using MPM to simulate a waterdrop falling onto the ground from different heights, there are around 10% failure cases (due to numerical instabilities), while a trained neural network is more stable; (2) **Unify different simulators and generalization to different materials**: we unified four materials in one unified network by creating data from both MPM and a rigid body simulator. If using an actual simulator, users would need to switch simulators for different materials. Moreover, due to our unified point representation, users do not need meshes to simulate rigid objects, while **standard rigid body simulator requires mesh as input**; (3) **Inference speed**: our base model and large model costs 1s and 3s for forward diffusion sampling, while MPM requires more than 4s. For one backward step, MPM requires 3min, while our method only needs 0.1s (no diffusion).
>
> > It is unclear how the model accounts for external elements like the ground plane if only the foreground object is segmented. For example, in Figure 1, how does the model determine that the penguin is supposed to collide with the table while the plane should stay in the air when rotating?
>
> **Our model takes floor condition as an additional condition (Line 172 of the paper) and the floor condition is calculated automatically from the input**. The model will automatically determine whether the points will collide with the boundary because the generated point trajectory is conditioned on the boundary. To calculate the floor condition, we use grounded segmentation or user input to select the floor and use VGGT to obtain the 3D points of the whole input image (including the floor and object), as our image-to-3D only reconstructs the foreground object. Then we align the coordinate systems of VGGT and our  image-to-3D using the correspondences of the foreground object. Finally, we obtain the value of the floor using principal component analysis based on the transformed ground plane points. If there is no floor detected, the value will be set to a placeholder.
>
> In Figure 1, we would like to clarify that the penguin won’t collide with the table because of the upward force. The plane case doesn’t contain a floor.
>
> > It is unclear whether the method would generalize to multi-object interactions.
>
> As acknowledged in the limitations section, modeling multi-object coupling or complex boundary conditions is beyond the scope of this work. To the best of our knowledge, no existing neural network approach currently addresses this challenge effectively. We did a preliminary experiment by creating a small dataset that has an object dragged towards a cube and colliding with the cube from different angles and distances. We trained our model on it and achieved 93.70% vIOU on the held out testing set. This preliminary experiment demonstrates the potential of our framework for multiple objects. Nevertheless, fully addressing this problem remains nontrivial and future work could explore incorporating stronger physics-informed losses to better handle complex deformations, as well as developing scalable and modular architectures that can flexibly support different numbers of interacting objects.
>
> > Comparison with Go-with-the-flow
>
> We run Go-with-the-Flow using the same test set and test protocol as in our paper. Following the official Github Repo, we use the cut-and-drag tool for motion control and then perform video generation. We observed that it supports only limited motion control, restricted to translation, rotation, and scaling of the original object, which hinders its ability to model complex deformations in elastic objects or morphological changes in materials like sand. Furthermore, it lacks support for 3D motion, such as the inward rotation in our UFO test case. These constraints result in Go-with-the-Flow performing worse than our method:
>
> |                                                | SA &uarr; | PC &uarr;  | VQ &uarr;  |
> |--------------------------------------------|------|------|------|
> | Ours                                        | 4.50 | 4.50 | 4.33 |
> | Go-with-the-flow                      | 3.58 |3.66| 3.58 |
>
> > Relevant papers.
>
> Thanks for pointing them out. We will add them in the revision.

---

> > ### Comment · Reviewer_F6p4 · 2025-08-05
> >
> > Thank you for your response.
> >
> > I have a follow-up question regarding the test case where the waterdrop impacts the ground. Did you use a fixed substep size for all frames in the MPM simulation? Typically, MPM simulators adopt adaptive time stepping based on the CFL condition to prevent numerical instability. Besides, the MPM simulator inherently handles multi-object interaction while the proposed neural simulator does not.
> >
> > Given this, I feel the major advantage of the proposed neural simulator lies in the speed. The MPM simulator usually requires small time steps and thus takes longer to simulate a stiffer material to avoid numerical explosion. I’m curious how the inference speed of the neural simulator compares to conventional MPM simulations across materials with varying Young's modulus.

---

> ### Comment · Area_Chair_dFw7 · 2025-08-05
> **Feedback Needed - Your AC**
>
> Dear Reviewer F6p4,
>
> I notice that the authors have submitted a rebuttal. Could you please let me know if the rebuttal addresses your concerns? Your engagement is crucial to this work.
>
> Thanks for your contribution to our community.
>
> Your AC

---

> ### Author Response · Authors · 2025-08-06
>
> We thank the reviewer for the in-depth comments and helpful discussions.
>
> > Substep size in Water Drop
>
> We use a fixed substep size $\Delta t=10^{-4}$ in the MPM simulation, a **conservative choice to avoid numerical instability following the common configuration in the widely used code of PhysGaussian [1]**. We sincerely thank reviewer for pointing out adaptive time stepping based on CFL condition. We admitted that choosing substep according to CFL can help prevent numerical instability. However, to our knowledge, CFL number is another hyperparameter and choosing an effective CFL number is not trivial, as studied by many research works such as [2], [3].
>
> **CFL check suggests that there is a tradeoff between robustness and speed of MPM.** In our waterdrop example, when using the common fixed substep size $\Delta t=10^{-4}$, the simulation will still fail in $10$% cases. After incorporating CFL check ($\max_ p \left\lVert\mathbf{v}_ p\right\rVert\le C_ {cfl}\frac{\Delta x}{\Delta t}$) and the fluid wave speed restriction $\Delta t \le C_{fluid} \frac{\Delta x}{\sqrt{\kappa/\rho}}$ (where we set $C_{cfl}=C_{fluid}=0.4$ following the convention) during MPM simulation, the substep size will shrink to $5 \times 10^{-5}$ for previous failure cases. The simulation can always success then but has lower speed, which takes more than $15$s to finish the simulation.
>
> > Handle Multiple-object interaction
>
> **Thanks to the point representation, our method is also inherently potential for multi-object interaction.** MPM achieves multi-object interaction by representing scene dynamics as point movement, and our model also predict point trajectories. Our previous response has shown our model's potential for multiple objects on a simplified setting. We will add that to the revised version. Future work might consider generate more diverse multi-object data, design better losses and architectures to fully address this problem, which is out of scope for this paper.
>
> > "Major advantage of the proposed neural simulator lies in the speed"
>
> **Our method has other advantages besides speed.** As mentioned in previous response, we have advantages of needing less hyperparameters, unifying other simulators, etc.
>
> > Inference speed across materials
>
> **Our method still has faster speed than MPM even when it uses adaptive time stepping based on CFL condition.** The below table shows the inference speed comparison of our model and MPM (we set the initial substep size to $\Delta t = 3 \times 10^{-3}$ and set the CFL condition to $\max_ p \left\lVert\mathbf{v}_ p\right\rVert$$\le 0.4\frac{\Delta x}{\Delta t}$ following [4], shrinking $\Delta t$ by 2 when violating the CFL condition) on elastic material with varying Young's modulus:
>
> | Young's Modulus| $10^{5}$ | $10^{6}$ | $10^{7}$ |
> | ------------------ | ---- | ---- | ---- |
> | Average Inference Time (MPM)| 0.22s | 0.69s | 1.49s |
> | Inference Time (Ours, 50 diffusion step)| 1.52s | 1.52s | 1.52s |
> | Inference Time (Ours, 25 diffusion step)| 0.76s | 0.76s | 0.76s |
> | Inference Time (Ours, 4 diffusion step) | **0.13s** | **0.13s** | **0.13s** |
>
> Since the uncertainty in our trajectory generation is relatively small, we are able to do 4-step inference, while still achieving strong performance. Thanks the reviewer for this question, which helps us demonstrate the speed advantage of our method.
>
> ||vIoU|CD|Corr|
> |------------------| ---- | ---- | ---- |
> |Inference Time (Ours, 50 diffusion step)|77.03%|0.0030|0.0016 |
> |Inference Time (Ours, 25 diffusion step)|77.22%|0.0029|0.0016 |
> |Inference Time (Ours, 4 diffusion step)|77.56%|0.0028| 0.0015 |
> |Inference Time (Ours, 1 diffusion step)|22.27%|0.0517| 0.0226 |
>
> **MPM benefits from techniques of numerical analysis, such as adaptive stepping based on CFL condition, while neural-based methods like our model can also be benefitting from many neural network accleration and diffusion model speeding up strategies**, such as 1-step diffusion distillation, model quantization etc. Besides, our model's backward speed is much faster, which is helpful for inverse problems.
>
> Once again, we sincerely thank the reviewer for the constructive feedback and the effort in improving this work. We hope these additional clarifications demonstrate the advantages and potentials of our method. We are also open to further suggestions and would be happy to incorporate additional improvements. We hope our responses address your concerns and would greatly appreciate your reconsideration of the overall evaluation.
>
> [1] Physgaussian: Physics-integrated 3d gaussians for generative dynamics, Xie et al.,
> [2] A precise critical time step formula for the explicit material point method, Ni et al.,
> [3] Effective time step restrictions for explicit MPM simulation, Sun et al.,
> [4] IQ-MPM: an interface quadrature material point method for non-sticky strongly two-way coupled nonlinear solids and fluids, Fang et al.

---

> > ### Comment · Reviewer_F6p4 · 2025-08-06
> >
> > Thanks for the detailed response and I am raising my rating to 4. I encourage the authors to include the discussed experimental results and analysis above to further strengthen the paper.

---

### Official Review · Reviewer_yGS5 · 2025-07-02

**Clarity:** 3
**Significance:** 3
**Originality:** 3
**Rating:** 5
**Confidence:** 4

**Summary:**

The paper introduces a method for an image-to-video generation with controllable physics. In a first step, the existing methods are used to lift the image to a 3D point-cloud. Given a force vector, a novel diffusion-based physics generation model estimates potential point trajectories. In the last step the generated trajectories a used to guide an existing image-to-video generation process.

**Questions:**

1. What are the advantages of a learned generative physics model over an actual simulator?
2. What are common failure cases of the method?
3. How did you settle on using $N=2048$ points for the point-cloud representation? This seems like a central design decision which is not further discussed in the paper. Especially for fine-grained sand simulation this seems like a surprisingly low value.

I am open to increase my score should the authors be able to show the benefit of generative dynamics over traditional simulation for the purpose of video generation and provide some insights into common failure cases.

**Ethical Concerns:**

["NO or VERY MINOR ethics concerns only"]

**Final Justification:**

After the rebuttal I decided to raise my score from "Borderline Accept" to "Accept" and to raise my confidence to a "confident". Especially the answers regarding the advantages of a learned dynamics model over traditional simulators are convincing and defused my main concern regarding the paper (W1/Q1). Furthermore, all my questions have been sufficiently answerd (Q1-Q3).

**Limitations:**

Yes

**Quality:**

3

**Strengths And Weaknesses:**

**Strengths**

1. **Qualitative results:** The shown qualitative results are of high quality and the generative object motions/deformation seem physically plausible
2. **Supplementary videos:** The supplementary video is well made and summarises the core parts of the method and showcases some good examples. The additional demo video on how to use the method is a nice cherry on top.
3. **Related work section:** The related works sections covers a wide range of different research directions while highlighting their shortcomings for the posed problem.
4. **Clarity**: The paper is well written and gives a clear and easy to follow introduction into the method and the experimental evaluation. The included figures sufficiently support the understanding of the method.

**Weaknesses**
1. **Why generative physics for video generation? (Also Question 1):** Physics generation and video generation are separate steps in the pipeline and the model requires explicit specification of the physics parameters so there is no direct need for using a generative method for the physics. Since the generative physics model seems to be the core contribution of the paper, it would be relevant to see how the full pipeline would perform if the learned model was replaced with a standard simulator. Alternatively, the authors could provide a more in-depth discussion on the necessity of using a learned simulator. (Also see Question 1)
2. **Failure cases:** The paper does not show any failure cases which would be relevant to give an actual impression of the models limitations. For example it would be relevant to see where the generative physics network breaks physical plausibility or controllability.

**Misc**
- Tab.1 seemingly contains a faulty looking value for DragAnything

---

> ### Author Rebuttal · Authors · 2025-07-31
>
> Thanks for your insightful questions, which is crucial for improving the quality of our work. The following are detailed responses to your concerns.
>
> > Advantages of a learned generative physics model over an actual simulator.
>
> Using neural simulator has several benefits: (1) **More robust**: we use a traditional simulator to generate training data by tuning their parameters and filtering failure cases. The filtered training data helps us train a robust neural simulator, even for cases where classical simulators struggle. For example, we found that when using MPM to simulate a waterdrop falling onto the ground from different heights, there are around 10% failure cases, mostly due to numerical instabilities, while a trained neural network is always stable; (2) **Easy-to-use**: our model requires minimal input from user, including only force direction and material. In contrast, MPM requires many other hyperparameters, including grid size, damping scale for grid velocity, frame dt and substep dt etc; (3) **Unifying different simulators and generalization to different materials**: we unified four materials in one unified network by creating data from both MPM and a rigid body simulator. If using an actual simulator, users would need to switch simulators for different materials. Moreover, due to our unified point representation, users do not need meshes to simulate rigid objects, while standard rigid body simulator requires mesh as input; (4) **Inference speed**: our base model and large model costs 1s and 3s for forward diffusion sampling, respectively, while MPM requires more than 4s. For the backward step, MPM requires 3min, while our method only needs 0.1s (no diffusion). Fast backward speed is especially beneficial for inverse problems.
>
> > Results of using a traditional simulator.
>
> We use the point trajectories from traditional simulators to generate videos on our testing set and use GPT for evaluation, the results are shown in the table below. Using our model is even slightly better than using the traditional simulator in certain metrics. The visual results mostly look similar. We will update these results in the paper. As such, using our model not only achieves great video generation results on successful cases of MPM, but also has the 4 benefits mentioned above.
>
> |                                                | SA &uarr; | PC &uarr;  | VQ &uarr;  |
> |------------------------------------------------|------|------|------|
> | Ours (2048 points, 4900 tracks)                             | 4.50 | 4.50 | 4.33 |
> | Use Traditional simulator (2048 points, 4900 tracks)  | 4.33 | 4.33 | 4.41 |
> | Use Traditional simulator (8192 points, 19600 tracks) | 4.33 | 4.33 | 4.17 |
>
> > Common failure cases of the method.
>
> (1) **Failure cases of our diffusion model**: we compared our results with MPM simulation on the Objaverse testing set. We found that cases with low vIoU typically involve thin structures or scenarios where our model predicts global motion but fails to accurately capture internal deformations. (2) **Failure cases due to video model**: we found in some cases, especially when the trajectories might conflict with the prior of video model, DaS cannot fully follow our trajectories. For example, when the user input for the object material is very stiff, but it appears soft according to RGB information. Then the resulting video might still be stiff. The video model might also hallucinate some unexpected content in the occluded region. For example, for an animal, it might generate five legs. (3) Failure cases due to image-to-3D is very rare.
>
> We will add these discussions and visualization of failure cases in the revision.
>
> > How did you settle on using N=2048 points for the point-cloud representation?
>
> We choose N=2048 considering that: (1) Prior works on 4D generation and reconstruction (PhysDreamer [1], SC-GS [2], MoSCA [3]) demonstrated that **real-world motions can be represented with a sparse number of basis or control points;** (2) The point trajectory is only providing a physics deformation signal for the video model and details will be generated by the video model itself. Our choice of 2048 points for the foreground object is further motivated by Table 3 in Diffusion as Shader (DaS), where the authors show that using 4900 points (including background static points) achieves similar results with more points (8100). In the Table above, we tested on simulated trajectories of more points as the input for the video model, results also show that using more points didn’t bring significant difference in performance.
>
> [1] PhysDreamer: Physics-Based Interaction with 3D Objects via Video Generation, Zhang et al., ECCV 2024
> [2] SC-GS: Sparse-Controlled Gaussian Splatting for Editable Dynamic Scenes, Huang et al., CVPR 2024
> [3] MoSca: Dynamic Gaussian Fusion from Casual Videos via 4D Motion Scaffolds, Lei et al., CVPR 2025
>
> > Misc: Tab.1 seemingly contains a faulty looking value for DragAnything
>
> Thanks for pointing this out. The number should be 2.8.

---

> > ### Comment · Reviewer_yGS5 · 2025-08-05
> >
> > Thank you for the extensive reply to my questions and concerns! Due to the clarity of the answers I have no follow-up questions.

---

> ### Comment · Area_Chair_dFw7 · 2025-08-05
> **Feedback Needed - Your AC**
>
> Dear Reviewer yGS5,
>
> I notice that the authors have submitted a rebuttal. Could you please let me know if the rebuttal addresses your concerns? Your engagement is crucial to this work.
>
> Thanks for your contribution to our community.
>
> Your AC

---

### Official Review · Reviewer_vXqE · 2025-07-03

**Clarity:** 2
**Significance:** 2
**Originality:** 3
**Rating:** 3
**Confidence:** 4

**Summary:**

PhysCtrl is a physics-grounded framework for controllable video generation, addressing existing models' lack of physical plausibility and 3D control. It features a diffusion-based generative physics network that learns physical dynamics for four materials, represented as 3D point trajectories. These trajectories act as control signals for video models, enabling explicit control over physics parameters and forces. Key contributions include its scalable framework, a diffusion model with spatiotemporal attention and physics-based constraints, and a large 550K synthetic dataset.

**Questions:**

1. How do the high costs and instabilities of physics simulators (for training data generation) impact the learned model's generalization or physical fidelity?
2. Does the fixed 2048-point representation limit expressive power or fidelity when handling complex geometries or fine deformations?
3. What are the specific limitations of the single-image 3D conversion step, and how do initial reconstruction errors affect subsequent physics simulation and video generation?

**Ethical Concerns:**

["NO or VERY MINOR ethics concerns only"]

**Final Justification:**

I have reviewed the authors’ rebuttal and will keep my score unchanged.

**Limitations:**

yes

**Quality:**

3

**Strengths And Weaknesses:**

PhysCtrl offers significant advantages for physics-grounded, controllable video generation. Its core is a diffusion-based generative physics network that learns 3D point trajectories for four material types. This approach circumvents the high computational cost and instabilities of classical physics simulators for inference, making it efficient. The framework incorporates a spatiotemporal attention mechanism and physics-based constraints, and is trained on a large 550K synthetic dataset. Experiments show it outperforms existing methods in visual quality and physical plausibility.
However, PhysCtrl has limitations. It currently focuses on single-object dynamics and only four material types, unable to model complex phenomena like multi-object coupling or intricate boundary conditions. While its inference is efficient, the training data relies on computationally expensive and potentially unstable physics simulators. Moreover, its experimental setup is highly resource-intensive (e.g., 8x Nvidia L40 for nearly two days per experiment), which prevented reporting error bars.

---

> ### Author Rebuttal · Authors · 2025-07-31
>
> Thank you for your insightful comments, which are crucial to improving the quality of our work. The following are detailed responses to your concerns.
>
> > Material types and modelling complex phenomena
>
> Our four material types (elastic, sand, rigid, plasticine) already cover a wide range of objects. **As acknowledged in the limitations section, modeling multi-object coupling or complex boundary conditions is beyond the scope of this work**. To the best of our knowledge, no existing neural network approach currently addresses this challenge effectively. However, since it was requested and to assess our framework’s potential in this setting, we conducted a preliminary experiment, where we created a small dataset that has an object dragged towards a cube and colliding with a cube from different angles and distances. We trained our model on it and achieved 93.70% vIOU, demonstrating the potential of our framework for multiple objects. Nevertheless, fully addressing this problem is nontrivial and future work could explore incorporating stronger physics-informed losses to better handle complex deformations, as well as developing scalable and modular architectures that can flexibly support different numbers of interacting objects.
>
> > Computational efficiency
>
> Compared to current AI models, the experimental setup of our trajectory generation model is not that resource-intensive. Our base/large model only needs 8L40 48G GPUs for training 30/80 hours, which is accessible in most academic settings. Our trajectory generation model also require less resources than our baselines for image-to-video generation: DragAnything needs A100s (training time not available) to directly finetune a much heavier video model (SVD); Wan2.1 and CogVideoX require at least hundreds of GPUs for training.
>
> > How do the high costs and instabilities of physics simulators (for training data generation) impact the learned model's generalization or physical fidelity?
>
> The high costs and instabilities are minor issues and have little effect on our model’s generalization and physical fidelity. Although the simulator runs slower than our model, we can utilize multiple low-end GPUs (e.g. 2080Ti) and generate data efficiently. The instability is mostly due to numerical issues, we filter them out by checking out of boundary values and NaN values. Our final dataset is high-quality, large-scale and contains diverse categories and geometries, which enables our trained model to be generalizable and stable. We will release the dataset.
>
> > Does the fixed 2048-point representation limit expressive power or fidelity when handling complex geometries or fine deformations?
>
> Prior work on 4D generation and reconstruction (PhysDreamer [1], SC-GS [2], MoSca [3]) has shown that **real-world motion can be effectively captured using sparse basis or control points**. Also, our choice of 2048 points for the foreground object is motivated by Table 3 in Diffusion as Shader (DaS), where the authors show that using 4900 points (including background static points) achieves similar results with more points (8100). From our observations, our method, trained on the diverse Objaverse dataset, performs reliably using 2048 sparse points, and we have not observed notable issues in fidelity or expressiveness. To further validate this, we tested using simulated trajectories of different numbers of points as the input for the video model. We follow our paper’s protocol to use GPT for evaluation and present the results in the table below. There is indeed minimal difference (GPT itself might also have some errors) when using different numbers of points to drive the video.
>
> We agree that sparse points may not model extremely fine deformations. However, in such cases, the video model itself might also not be able to generate those delicate details. To the best of our knowledge, no existing methods for physics-grounded video generation are capable of handling such fine-grained deformations effectively.  We will discuss this challenge for all the existing methods (including our work) in the limitation and leave this for future work.
>
> |                                                | SA &uarr; | PC &uarr;  | VQ &uarr;  |
> |------------------------------------------------|------|------|------|
> | Ours (2048 points, 4900 tracks)                             | 4.50 | 4.50 | 4.33 |
> | Use Traditional simulator (2048 points, 4900 tracks)  | 4.33 | 4.33 | 4.41 |
> | Use Traditional simulator (8192 points, 19600 tracks) | 4.33 | 4.33 | 4.17 |
>
> [1] PhysDreamer: Physics-Based Interaction with 3D Objects via Video Generation, Zhang et al., ECCV 2024
> [2] SC-GS: Sparse-Controlled Gaussian Splatting for Editable Dynamic Scenes, Huang et al., CVPR 2024
> [3] MoSca: Dynamic Gaussian Fusion from Casual Videos via 4D Motion Scaffolds, Lei et al., CVPR 2025
>
> > Limitations of single image to 3D and its errors on subsequent steps
>
> The single-image-to-3D step produces reasonable geometry overall and is very unlikely to yield severely distorted or implausible shapes. While minor artifacts such as surface noise may occur, they have minimal impact on subsequent steps. We achieve such robustness to geometric variations because we trained the trajectory generation network on the diverse Objaverse dataset and applied surface noise augmentation during data generation. Moreover, the predicted point trajectories are used as motion signals to guide the video model, and the video model will generate fine visual details with this guidance.

---

> ### Comment · Area_Chair_dFw7 · 2025-08-05
> **Feedback Needed - Your AC**
>
> Dear Reviewer vXqE,
>
> I notice that the authors have submitted a rebuttal. Could you please let me know if the rebuttal addresses your concerns? Your engagement is crucial to this work.
>
> Thanks for your contribution to our community.
>
> Your AC

---

> ### Author Response · Authors · 2025-08-06
>
> Dear Reviewer vXqE,
>
> Thank you for your thoughtful and constructive reviews. We have submitted our rebuttal and hope it addresses your concerns effectively. We would greatly appreciate it if you could join the discussion and share any further thoughts or questions.
>
> We are happy to provide additional clarifications or details as needed and are committed to improving the paper based on your feedback.
>
> Best regards,
> The Authors

---

### Author Response · Authors · 2025-08-02

Dear Reviewers,

Thank you again for your thoughtful and constructive review! We hope our rebuttal has addressed your concerns. As the discussion period is short, we’d really appreciate it if you could let us know whether our responses have resolved your questions. If there are any remaining concerns or suggestions, we’d be very grateful for the chance to further clarify — we’ll do our best to improve the paper accordingly.

Best regards,

The Authors

---

### Author Response · Authors · 2025-08-05

Dear Reviewers:

Thank you for your efforts in reviewing our paper! Your feedback is valuable to us and we greatly appreciate your time and insights. We have submitted our rebuttal and tried to address your concerns as thoroughly as possible. As the discussion period is quickly passing, we’d really appreciate it if you could let us know whether our responses have resolved your questions. We would be grateful for any additional feedback to help improve the paper and clarify any remaining points.

Best Regards,

Neurips submission #2097 Authors

---

### Note · Authors · 2025-08-12

We sincerely thank AC’s support and the constructive feedback of all reviewers. We are encouraged by their recognition of the key contributions and strengths of our work. For clarity and simplicity, we refer Reviewers vXqE, yGS5, F6p4, eLwH as R1, R2, R3 and R4 in this response.

In particular, we appreciate the acknowledgement of our method: a scalable framework (**R1**), efficient inference compared to classical simulators (**R1**), a compressive and technically sound pipeline (**R3**), novel and interesting techniques (**R4**) with clear evaluation metrics (**R4**). We are pleased that reviewers generally recognize our results: physics-grounded and controllable video generation (**R1**), outperforms existing methods in visual quality and physical plausibility (**R1**), high-quality and physically plausible qualitative results (**R2**), impressive results and significant improvement over baselines (**R3**), clearly superior compared with baselines (**R4**). We also appreciate the reviewers’ comments on our presentation, noting that our paper is well-written and easy-to-follow (**R2, R3**), has clear writing, motivation, and excellent organization (**R4**), with good video demonstration (**R2, R3**).

During the rebuttal, we mainly discussed the failure cases of our model (**R1, R2, R4**), made clarifications of using 2048 points (**R1, R2**), further clarified the benefits of using a neural simulator (**R2, R3**), discussed extension to multiple objects (**R1, R3**) (although this is out of scope and also not well studied by prior works), provided additional comparisons (**R3**), and more explanations and details of the method and results (**R3, R4**).

*While we did not receive further input from R1 during the discussion period, we still hope our rebuttal has addressed their concerns effectively*. The points they raised—modeling multiple objects, using 2048 points, and the limitations of image-to-3D—overlap with questions from other reviewers. We also provided comprehensive clarifications on computational efficiency and training data generation.

We are grateful for all reviewers’ constructive feedback, which have helped us strengthen our work. We will incorporate these discussions into our revision. Thank you!

---

### Decision · Program_Chairs · 2025-09-17

**Decision:**

Accept (poster)

**Comment:**

The submission received divergent ratings, including 1 Borderline reject, 1 Borderline accept, and 2 Accept. Three reviewers were actively engaged with the discussion and gave positive scores. Reviewer vXqE, who gave the only negative score, expressed concerns about the generalization. The authors and the AC tried to engage Reviewer vXqE in the discussion. Reviewer vXqE acknowledged that the rebuttal addressed the concerns well, but without changing the rating.

After checking the review, the rebuttal, the manuscript, and the discussion, the AC concurs with the other three reviewers' assessment and recommends acceptance. The work has a decent contribution to video generation and deserves sharing with the community.